# Control of Synthetic Hairpin Vortices in Laminar Boundary Layer for Skin-Friction Reduction

**Bonguk Koo** [1] and **Yong-Duck Kang** [2,*]

1   Department of Naval Architecture and Marine Engineering, Changwon National University,
    20 Changwondaehak-ro, Uichang-gu, Changwon-si, Gyeongsangnam-do 51140, Korea;
    bonguk9@changwon.ac.kr
2   Department of Naval Architecture and Ocean Engineering, Dong-Eui University, 176 Eomgwangro,
    Busanjin-gu, Busan 47430, Korea
*   Correspondence: ydkang@deu.ac.kr; Tel.: +82-(051)-890-2815

**Abstract:** The results of flow visualization and hot-film measurement in a water channel are presented in this paper, in which the effectiveness of controlling synthetic hairpin vortices in the laminar boundary layer is examined to reduce skin friction. In this study, hairpin vortices were generated by periodically injecting vortex rings into a cross flow through a hole on a flat plate. To control the hairpin vortices, jets were issued from a nozzle directly onto the head of the hairpins. The results of the flow visualization demonstrated that the jets destroyed the hairpins by disconnecting the heads from their legs, after which the weakened hairpin vortices could not develop. Therefore, the circulation around the legs was reduced, which suggests that the direct intervention on the hairpin heads resulted in the reduction of streamwise stretching. Data obtained by a hot-film sensor showed that the high-speed regions outside the hairpin legs were reduced in speed by this control technique, leading to a decrease in the associated local skin friction.

**Keywords:** fuel efficiency; frictional resistance reduction; hull-fluid interaction; hairpin vortex; jet control

---

## 1. Introduction

There is more attention to fuel efficiency of ships by national and international organizations due to increasing environmental pollution [1]. Therefore, various research topics have been proposed to reduce the frictional resistance that increases squarely with the ship speed. Reducing frictional resistance includes decreasing the wetted surface area, the ship speed, and the interaction between hull surface and fluid. However, the changes in the principal dimensions are risky and increase costs, therefore many researchers have tried to understand the mechanism of energy regeneration in the process of hull surface interaction with vortices generated in the boundary layer. Most authors have investigated the importance of streamwise vortical structures in the production of skin friction with a view to manipulating them to reduce drag [2–10].

It has been believed that coherent structures produce turbulent energy that is key to their ability to sustain themselves in boundary layers. Kline et al. [11] showed that the pumping motions within the viscous sublayer formed low-speed streaks by inducing a vortex pair, which interacted with the outer layer through burst events. Kim et al. [12] followed this work with flow visualization and hot-wire measurements, and observed the generation of highly intermittent turbulence near the wall, which became concentrated during burst events. Wallace et al. [13] showed that two main motions, the ejection-type motion with a negative streamwise velocity ($u$) and positive wall-normal velocity ($v$) and the sweep-type motion with positive streamwise velocity and negative wall-normal velocity,

contributed significantly to the Reynolds stress in the wall region. Kim [14] reported in his LES results that counter-rotating streamwise vortices were created as a result of "splatting" and also speculated that this pair of counter-rotating vortices could be the two legs of a hairpin vortex.

The hairpin-vortex hypothesis first proposed by Theodorsen [15] was well documented by Head et al. [16] in their flow visualization studies over a range of Reynolds numbers. The vortices were elongated and developed into more hairpin-vortex shapes at a moderate Reynolds number and large-scale structures consisting of several vortices. Robinson [17] summarized this topic by reporting turbulent structures that consisted of hairpin legs in the buffer and log regions, whereas arches, necks, and the heads of hairpins occurred in the log and wake regions. More details of the development of hairpin vortexes were described by Smith et al. [18], both experimentally and computationally, who suggested that once formed, a vortex loop moved away from the wall by self-induction and downstream due to the background shear flow. The trailing legs of the loop remained in the wall region and were stretched to form a pair of streamwise vortices. Low-speed fluids accumulated between the legs and acted to pump those fluids away from the wall. Dhanak et al. [19] showed a highly simplified model that grew into a hairpin vortex in a two-dimensional simulation. As the head of the disturbed line vortex moved away from the surface, it developed into a horse-shoe shaped vortex and either side stretched toward the wall to form legs. Adrian [20] explained the development of a hairpin vortex as reaching around the edge of the boundary layer, with the head of the hairpin vortex moving faster at a higher mean velocity than the legs, which stayed near the wall. Thus, the legs underwent an intensification process resulting in the growth of the hairpin vortex toward the edge. The streamwise vortices forming at the near-wall region were parts of large-scale structures in the boundary layer. It is commonly believed that the downwash side of streamwise vortices induces high-speed fluid toward the wall, called a sweep event, whereas low-speed fluid moves away from the wall, which is called an ejection event. Kravchenko et al. [21] showed that high frictional drag was closely correlated with the sweep side of streamwise vortices. Furthermore, the authors considered that skin friction was increased when high-speed fluid was transported to the wall.

Vortical structures, which are responsible for about half of the turbulence energy at the near-wall region, have been considered to be sustained by a cycle without the influence of outer flow [22,23]. Therefore, flow control methods have usually focused on the modification of near-wall structures. Although the outer flow did not stimulate the formation of near-wall structures, based on the analysis of Jimenez et al. [24], it did modulate their lengths with respect to the boundary layer. As the head of the hairpin vortex reached the edge of the boundary layer, the size of the vortex core of the legs was reduced and the vorticity was increased. This top-down process, wherein large-scale turbulence structures influence small-scale near-wall turbulence structures, could contribute to skin-friction drag. It was proposed that the superstructures not only superimposed a low wavenumber onto near-wall structures as a footprint, but they also interacted more progressively with small-scale motions as the Reynolds number increased [25–28]. Moreover, it was also suggested that the low wavenumber of the wall shear was due to the presence of the large-scale structures [29–33].

Based on the influence of streamwise vortices, many investigations regarding skin-friction reduction in turbulent boundary layers have directly targeted near-wall events [2,3]. As a passive control device without any energy input [4–6], riblets on the surface were used to manipulate near-wall structures. The researchers reported that drag reduction could be obtained when longitudinal vortices interacted to trigger the burst event, where closely spaced riblets reduced the amount of exposed area and restricted spanwise movement. Alternatively, active techniques that require energy input have been used to maximize the net drag reduction in terms of control efficiency. Du and Karniadakis [7] reported in their direct numerical simulation (DNS) by Lorentz force that a transverse traveling wave was more efficient than riblets for spanwise oscillation control. They showed that a traveling wave could remove alternating streamwise streaks and hairpin vortices. Choi et al. [8] carried out DNS to reduce skin friction by blowing and suction, depending on either sweep or ejection. They reported the mechanisms of drag reduction as occurring in two ways; first, active control hindered the sweep motions that

bring high-speed velocity to the wall and, secondly, it deterred the regeneration of secondary vortices by preventing ejection motions. In references [9,10], they attempted real-time opposition control by wall-normal jets at the wall to cancel high-speed velocity toward the wall. They selectively counteracted sweep events at the near-wall region and obtained skin-friction reduction.

Few methods have been proposed to reduce skin friction in the turbulent boundary layer by controlling the large-scale structures, although this approach would last longer and cover a wider area than wall-based controls with high Reynolds numbers [34,35]. Large-eddy break-up (LEBU) devices in the outer part of a boundary layer interrupt the energy production loop by directly interacting with the large-eddy structures, thereby weakening burst events near the wall and reducing skin-friction [36–40]. As the shedding wake behind the LEBU prevents large-scale structures from moving toward the wall, the events producing Reynolds stress at the wall could be reduced. Hutchins and Choi [41] used an array of thin vertical blade devices (VBDs) in the turbulent boundary layer to reduce local skin friction by up to 30% downstream of the blades. The authors explained that when large-eddy structures passed through the VBD array, their shoulders were sliced off and the link between the outer and inner structures was disconnected, thereby reducing skin-friction drag.

Large-scale structures are recognized as the main features affecting turbulence production and dissipation through the boundary layer during its growth process. The aim of the present work was to investigate the effectiveness of an outer control strategy, whereby a direct intervention on the hairpin vortices is performed by control jets located outside the boundary layer to decrease the vorticity around the legs of the hairpin vortex and reduce the wall shear stress. The outer layer-based technique proposed in this study is not applicable in engineering purposes. However, as a part of a feasibility study for this ambitious attempt, the experiments have been carried out in a laminar boundary layer since the flow phenomenon is less complicated than that of a turbulent flow. It would have an additional advantage of manipulating energetic near-wall structures on top of the effect on large-scale structures. A detailed description of the initial stage of this study regarding the direct intervention on hairpin structures in the turbulent boundary layer can be found in Kang et al. [42]. This article presents how the flow with hairpin vortices was simulated in a laminar boundary layer. Then, a direct intervention of hairpin vortices was applied to reduce skin-friction drag. Both flow visualization and hot-film measurements were made to confirm this control methodology.

## 2. Experimental Set-Up

### 2.1. Water Channel with Dye System

This study was conducted in an open water channel 7.3 m long with a cross section of 0.62 m × 0.30 m (width × height). We suspended a 4.0 m long flat plate with 20:1 elliptic leading edge made of Perspex 0.1 m from the bottom of the water channel, as shown in Figure 1. A 0.25 m long flap was attached to the trailing edge of the test plate and set to 17° to get zero pressure gradient over a test plate. The flap angle was adjusted to separate fluid evenly into upper and lower at the leading edge. The streamwise direction was along the $x$-axis, the wall-normal direction was along the $y$-axis, and the spanwise direction was along the $z$-axis.

Unlike a hot-wire probe, a hot-film sensor with an overheat ratio of 1.05 is very sensitive to temperature, so temperature control was very important during this experiment. The heat generated by the high-speed pump blades caused the water temperature to rise by 0.1 °C within 10 min. Thus, a heat exchanger made of copper pipe was installed in the upstream tank to maintain a constant water temperature by the passage of cold tap water.

A 5 mm hole on the test plate was drilled 0.37 m from the leading edge, through which red dye was supplied to generate a vortex ring. To prevent disturbance inside the laminar boundary layer, a 0.25 mm jet nozzle was installed well outside the boundary layer at $y/\delta = 1.67$, which was connected to a green dye tank. Each tube was connected to a 10.0 L plastic tank with a spigot, in which the same water level was maintained to maintain the same pressure.

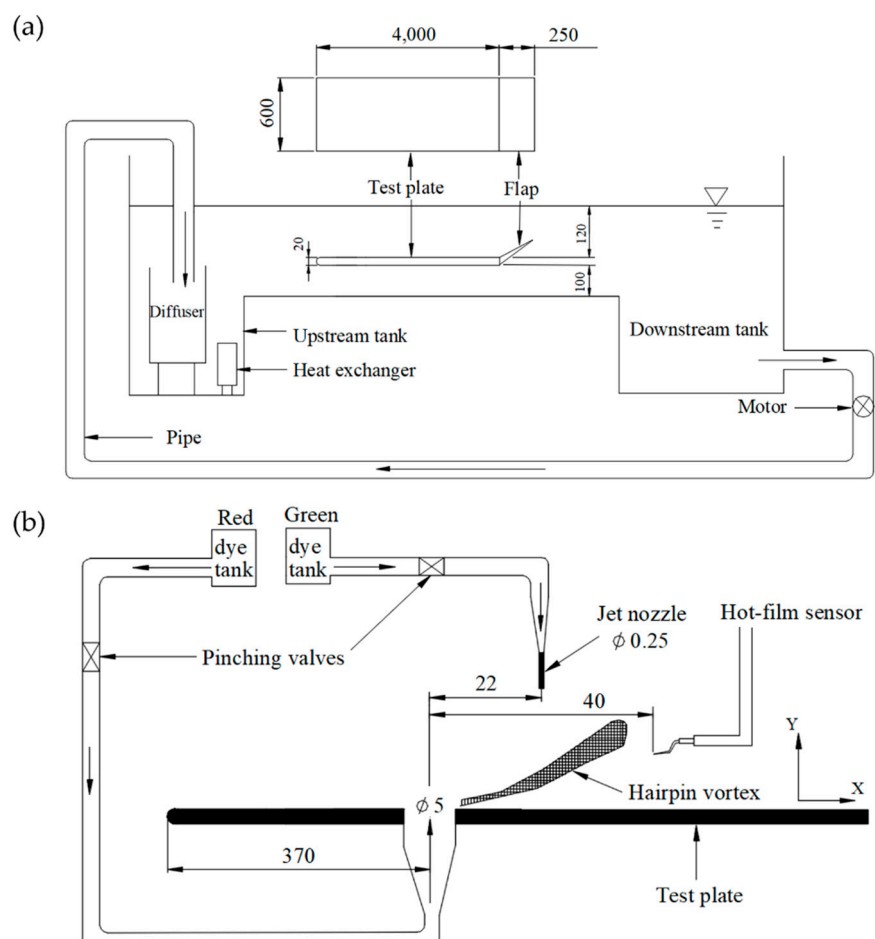

**Figure 1.** Experimental configurations: (**a**) water channel with test plate; (**b**) hairpin vortex control by jet with hot-film measurement.

## 2.2. Sensor Calibration and Velocity Measurement

The data were obtained using a Dantec 56C Constant Temperature Anemometry (CTA) system with a 55R15 hot-film sensor (Dantec Dynamics, Skovlunde, Denmark). This 3 mm long probe has a sensor length of 1.25 mm, which was operated at an overheat ratio of 1.05. An automatic traverse system was used to move the probe around in the YZ plane, which was controlled by a stepper motor. The traverse resolution was 6.25 μm per step in the wall-normal direction with an accuracy of ±300 μm. The hot-film sensor measured the streamwise velocity in the YZ plane at $x = 41$ cm from the leading edge. Velocity measurements were performed at 17 spanwise positions in the range of −8 mm ≤ z ≤ 8 mm with a 1.0 mm interval. In addition, a wall-normal velocity scan was performed at 36 positions with a 0.1 mm interval from the wall to 1.5 mm, a 0.5 mm interval to 7 mm, and a 1.0 mm interval to 17 mm.

To calibrate the hot-film sensor, it was placed in the free-stream region of the water channel. The output voltage of the sensor was sampled at 100 Hz for 60 s and was referenced to the indicated velocity of a Nixon 403 Streamflo Velocity Meter (Nixon Flowmeters Ltd, Cheltenham, UK). A vane-type anemometer measured the velocity range from 2.5 cm/s to 1.5 m/s with an accuracy of ±2%. However, the Nixon velocity meter had difficulty operating below the minimum measured velocity.

For this reason, the hot-film sensor was traversed through the channel by the stepper motor in still water. The stepper motor was initially accelerated and kept at constant speed before decelerating at the end. Voltage measurements were obtained during the constant-speed portion for comparison with the known velocity of the motor. The sensor on the motor was moved through the channel several times at

different speeds up to 5 cm/s. The calibration velocities at an ambient temperature were averaged and plotted against the average voltage, as shown in Figure 2. Based on the sensor voltages against the referenced velocities, calibration curves were fitted with the fourth-order polynomial.

The temperature was monitored using a platinum resistance thermometer (PRT) with an accuracy of ±0.01 °C. The temperature drift in our experiment was ±0.2 °C. In practice, it was impossible to generate calibration curves for all temperatures, so to process the velocity data, linear interpolation was used to correct the temperature effect on the calibration curve. The free-stream velocity drift was also corrected by constantly monitoring the Nixon velocity meter.

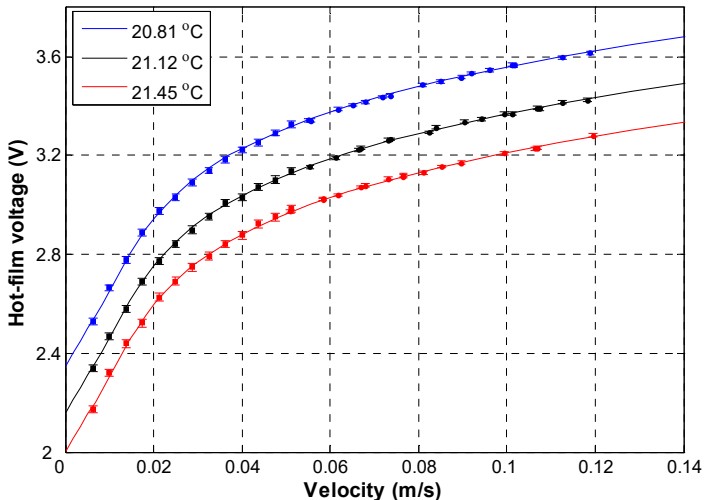

**Figure 2.** Hot-film calibration at three temperatures.

The boundary layer over the test plate remained laminar up to at least the measurement location, where the boundary layer thickness δ was about 9 mm. The Reynolds number based on the distance from the leading edge was $Re_x = U_\infty \cdot x / \nu = 4.1 \times 10^4$, where $x$ is the distance from the leading edge (0.41 m), $U_\infty$ is the free-stream velocity (0.10 m/s), and $\nu$ is the kinematic viscosity of the water ($1.005 \times 10^{-6}$ m$^2$/s). The nondimensional velocity profile of the laminar boundary layer showed good agreement with the Blasius profile [43], as illustrated in Figure 3.

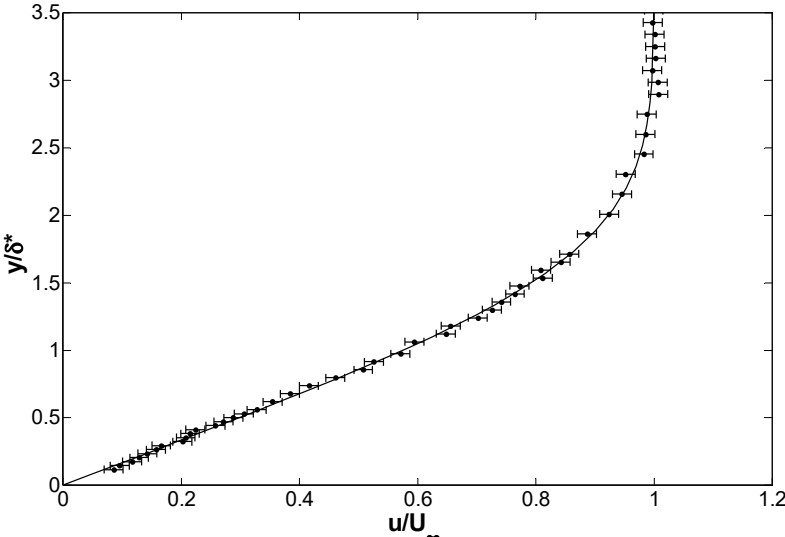

**Figure 3.** Comparison of the experimental data (closed circles) with the Blasius profile (black line). Error bar is indicated at data points.

### 2.3. Hairpin Vortex Generation and Jet Control

Figure 4 shows the velocity change of the control jets at the nozzle exit and that above the vortex ring in the still water. The locations of the hot-film sensor for measuring the velocity were in the center of the hole and at the jet nozzle exit. In creating vortex rings from the test plate through the 5 mm hole, a Sirai S105-06 solenoid pinch valve was actuated by a Wave Factory 1976 signal generator.

The valve was a two-way, normal-closure-type valve with a 650 g pinching strength. The pinch valve opened a 5 mm clear C-FLEX® tube (SAINT-GOBAIN, Charny, France) when the signal generator generated a square wave with a 90 ms duration at a frequency of 1 Hz. The red dye initially formed a vortex ring at the exit of the hole, and then grew into a hairpin vortex with cross flow, as shown in Figure 5. This produced the same vorticity distribution around the hairpin structure in the turbulent boundary layer. An identical solenoid pinch-valve system was used to issue control jets toward the hairpin vortices. The valve was opened for 150 ms to issue jets through a nozzle to counteract the velocity field at the head of the hairpin vortices. The jet velocity was $u_{JET} = 0.017$ m/s and the pulsing wall-normal velocity for creating vortex rings was $u_{HV} = 0.22$ m/s in still water.

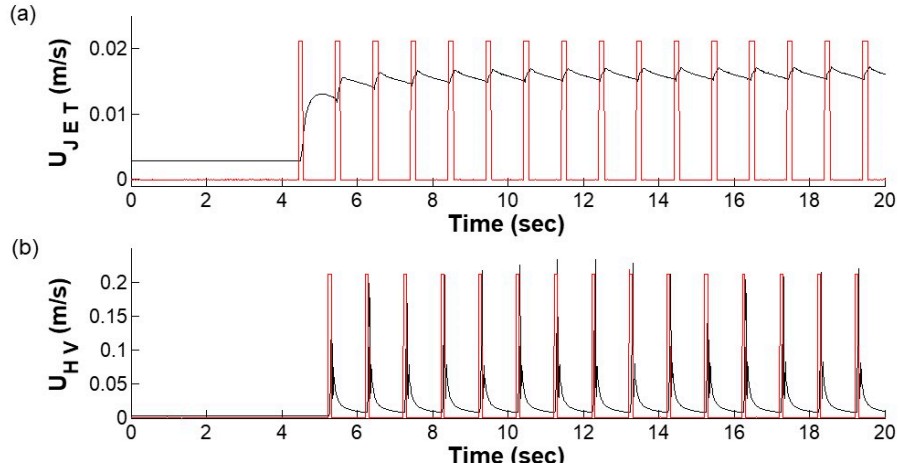

**Figure 4.** Velocity changes of (**a**) jet ($u_{Jet}$) and (**b**) hairpin vortex ($u_{HV}$) in still water. The solid line shows the velocity change and the dotted line shows the timing of the pinch valve.

Figure 5 shows a sequence of the development of a hairpin vortex from a vortex ring. When the vortex ring is pumped into the cross flow, outbound vorticity is generated around the ring. The arrows in the figure indicate the vorticity distributions around the hairpin vortex. However, the upstream part of the vortex ring is canceled by the incoming flow [44], which results in a reduction of the self-induced velocity. In contrast, the downstream part of the vortex ring develops into the head of a hairpin vortex as it is convected by the cross flow. This figure also shows the streamwise stretching of the side of the vortex ring becoming hairpin legs. Once the downstream part of the ring has moved away from the wall, it convects faster than the upstream part.

Chang and Vakili [45] and Sau and Mahesh [46] studied the dynamics and trajectories of vortex rings in a uniform cross flow to increase mixing. They concluded that no considerable interaction occurred between neighboring vortex rings below a pinching frequency of 4 Hz for a jet-to-crossflow velocity ratio of 3. In this study, vortex rings were introduced at a wall-normal velocity of 0.22 m/s into a boundary layer with a free-stream velocity of 0.10 m/s. To minimize interference between successive rings, the jet-to-crossflow velocity ratio was 2.2 and the frequency of producing vortex rings was 1 Hz.

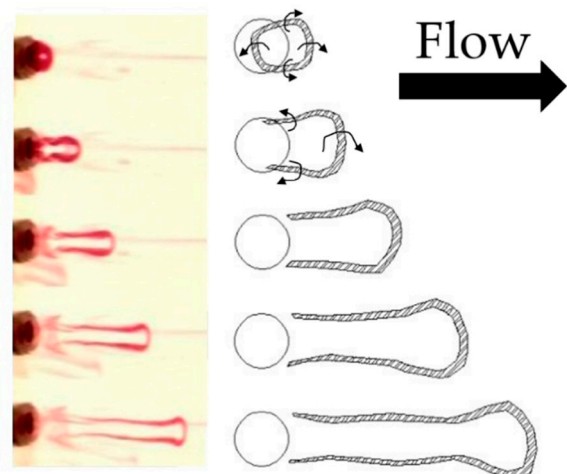

**Figure 5.** Development of hairpin vortex from vortex ring issued through test plate in cross flow. Arrows indicate vorticity distributions around the hairpin vortex.

*2.4. Vortex Control Methodology*

Figure 6 illustrates the concept of direct intervention on hairpin vortices. The black square at the top of the figure is the jet nozzle, and the gray arrow indicates the direction of the control jet onto the head of the hairpin vortex. The black arrows around the hairpin vortex indicate the vorticity distribution. The hot-film sensor is located at the right top corner. When the vortex ring is introduced into the boundary layer, an outbound vorticity forms around the ring. This vorticity distribution causes a Kutta–Joukowski lift (Magnus effect), such that the downstream part of the vortex ring tilts upwards [44,45]. First, the upwash of low-speed velocity in the center of the hairpin vortex is hit by the control jet. Here, we set the jet velocity to be equal in magnitude but opposite in direction. The impingement of the control jet on the hairpin vortex disconnects the heads from their structures and detaches them from the rest of the vortical structures. By detaching the heads, the streamwise stretching of the hairpin legs is reduced, leading to a reduction in vortex circulation. This, in turn, reduces the downwash of high-momentum fluid from the velocity induced outside the hairpin legs.

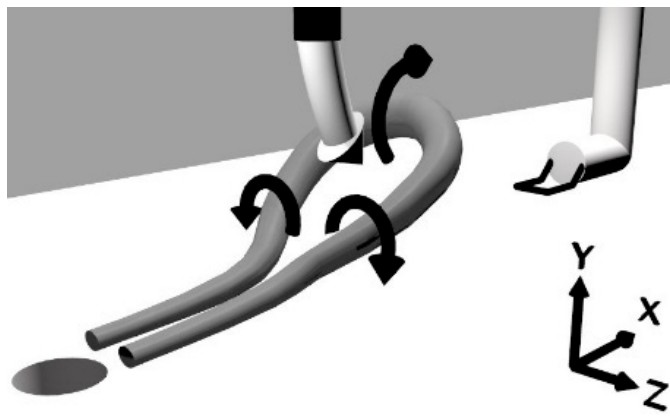

**Figure 6.** Conceptual illustration of the control strategy.

## 3. Results and Discussion

As a part of a feasibility study for an ambitious attempt, flow visualization and hot-film measurement were performed at the laminar boundary to verify that controlling large-scale structures could cause a cascade of changes in small-scale structures at the near-wall region. In the control of large structures, a periodic series of hairpin vortices is needed to achieve precisely timed control. However,

an experimental configuration with a hemisphere [47,48] or fluid injection through a flat plate [49–51] has produced hairpin vortex structures with jitters.

Thus, we introduced a new system wherein a vortex ring was generated through a hole on a flat plate, which developed into a hairpin vortex with a cross flow, as shown in Figure 7, where the left column shows the no-control case and the right the control case.

The solid line at the top of each photograph in the figure indicates the jet nozzle and the dotted line in Figure 7h shows the measurement position at $x = 4.4\delta$, where the hot-film sensor scanned the streamwise velocities in the YZ plane. The time interval between successive photos was 0.12 s. The downstream part of the vortex ring developed into the head of a hairpin vortex toward the boundary edge as it was being convected by the cross flow at the initial wall-normal velocity. In contrast to this, the upstream part of the ring vortex was weakened by the interaction with the incoming flow in the opposite direction and remained near the wall region. Once the downstream part of the ring moved away from the wall, it convected faster than the upstream part due to the higher mean velocity. As a result, the upstream part experienced a stretching and intensification process. Then, the control jet hit the head of the hairpin vortex, as shown on the right side of Figure 7d, and the interaction between the control jet and hairpin vortex continued afterwards. This flow visualization shown in Figure 7f–h demonstrates that the head of the hairpin vortex was being disconnected from the rest of structure.

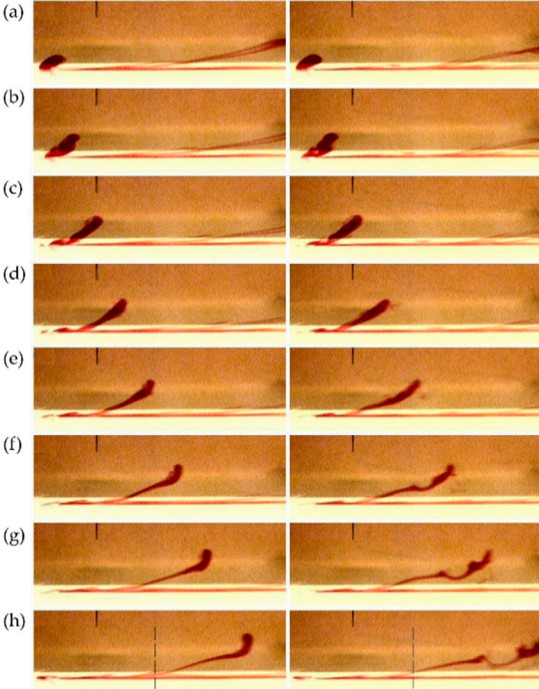

**Figure 7.** Time series of the flow visualization of the no-control case (left column) and that with control (right column) at $x = 4.4\delta$ indicated in (**h**), with the dotted line indicating the position of the hot-film sensor for scanning the YZ plane. (**a**) vortex ring created on the left corner; (**b–d**) hairpin vortex development reaching to boundary edge; (**e–h**) streamwise stretching process of legs.

Figure 8 shows the phase-locked averaged velocity fluctuations in the YZ plane at $x = 4.4\delta$, with the left column showing the no-control case and the right column the control case. The time interval between successive photos was 0.05 s.

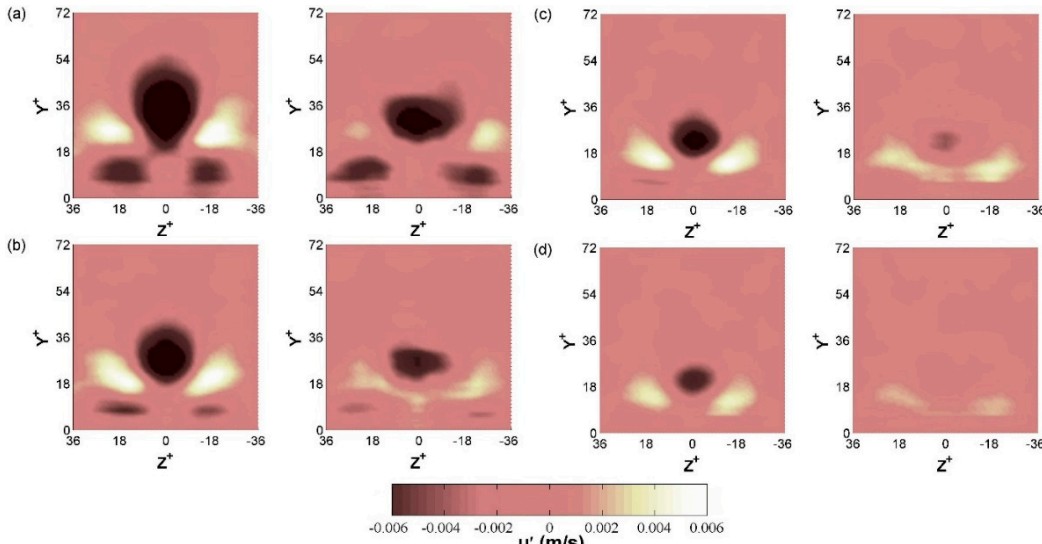

**Figure 8.** Phase-locked averaged velocity fluctuations of the YZ plane at *x* = 4.4δ, as indicated by the dotted line in the last figure of Figure 7. Induced high (white) and low (black) momentum towards near-wall: (**a,b**) by vorticity of neck part; (**c,d**) by vorticity of leg part.

In Figure 8a, the white patches indicate the increased velocity fluctuations caused by the wallward (sweep) motion outside the hairpin vortex, and the black patches indicate the decreased velocity fluctuations by the upwash (ejection) motion inside the hairpin vortex. As time elapsed, the vortex legs passed through the measurement position so that their high velocities were approaching the wall, thereby increasing the skin friction. The measured velocity fluctuations correspond to those in Figure 7e, although the visualization results indicate the hairpin vortex structure itself with dye. Therefore, the hairpin vortex structure would exist between the low and high velocity fluctuations. In Figure 8d, it can be clearly seen that as soon as the control jet hit the head of the hairpin vortex, both the low- and high-velocity fluctuations were reduced and this also affected the legs as shown. We consider that the jet stopped the stretching and intensification of the legs by breaking the link between the legs and the head, thereby reducing the velocity fluctuations.

Ensemble-averaged velocity fluctuations were used to reconstruct a three-dimensional hairpin vortex. Figure 9 shows the isosurfaces of streamwise velocity fluctuations, wherein velocities higher than the mean velocity are in red and lower velocities are in blue. Red color indicates the isosurface of $u/U_\infty = 2.7\%$ and blue color indicates that of $u/U_\infty = -8.2\%$.

In Figure 9a, the hairpin vortex colored dark gray is located between the low-speed and high-speed velocities. The hairpin vortex drew the low-speed fluids from the wall by pumping between its legs, while inducing high-speed fluids around the hairpin vortex. Here, the hairpin vortex structure can be detected by a marker function defined by Equation (1) as follows:

$$\Psi_{hp} = \sqrt{\left(\frac{\partial \langle u \rangle}{\partial z}\right)^2 + \left(\frac{\partial \langle u \rangle}{\partial y}\right)^2} \tag{1}$$

where $\langle u \rangle$ denotes the ensemble-averaged streamwise velocity fluctuation.

Thus, the detected vortical structure was revealed with the marker function $\Psi_{hp} = 420$.

Hutchins [52] devised the marker function $\Psi_{hp}$ by squaring the vorticity components in terms of the spatial gradient of the streamwise velocity. However, this marker function only works in and around near-wall turbulent structures, where the streamwise velocity *u* is strongly correlated with the negative wall-normal velocity −*v*. The streamwise vorticity is defined as $\omega_x = \left(\frac{\partial w}{\partial y} - \frac{\partial v}{\partial z}\right)$, where *w* is spanwise velocity. This does not involve the streamwise velocity. However, ejection (−*u* and +*v*) and sweep (+*u* and −*v*) events are highly correlated in the near-wall region, so that wall-normal velocity is

replaced by streamwise velocity, where streamwise vortices are predominant. Thus, the streamwise vorticity can be simplified as $\omega_x = (\frac{\partial w}{\partial y} - \frac{\partial(-u)}{\partial z}) \approx \frac{\partial u}{\partial z}$. In the same process, wall-normal vorticity can be given as $\omega_y = (\frac{\partial u}{\partial z} - \frac{\partial w}{\partial x}) \approx \frac{\partial u}{\partial z}$ as well as spanwise vorticity $\omega_z = (\frac{\partial v}{\partial x} - \frac{\partial u}{\partial y}) \approx -\frac{\partial u}{\partial y}$.

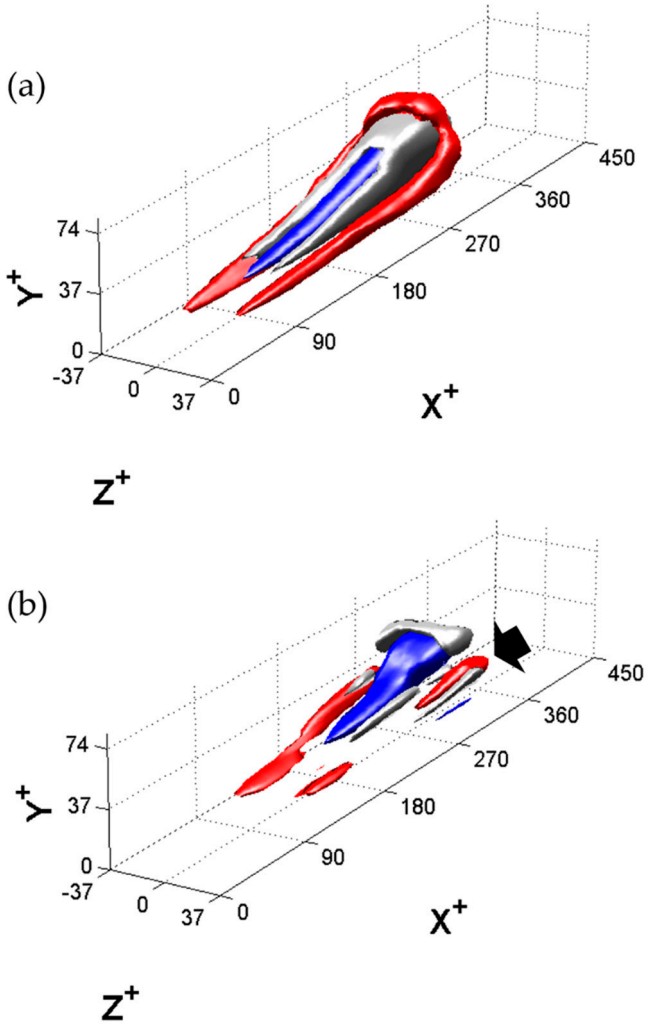

**Figure 9.** Isosurfaces of ensemble averaged streamwise velocity fluctuations. (**a**) No control and (**b**) control.

We obtained the streamwise length from the time-series data using Taylor's frozen-flow hypothesis by taking the free-stream velocity as the convection velocity. The length scales were normalized by the viscous wall unit, $l^+ = l \cdot u_\tau / \nu$, with $u_\tau = 0.0045$ m/s. The hairpin vortex was $x^+ \approx 200$ long, $y^+ \approx 60$ high, and $z^+ \approx 25$ wide, which is similar to the vortex size shown in Figure 7. With the control in Figure 9, the control jet counteracted and weakened the upwash motion within the hairpin vortex. The area of low-speed velocity was reduced up to 40% compared with the no-control case and the high-speed area was also decreased. The red patch pointed by a black arrow in Figure 9b was caused by the penetration of the jet toward the wall, because the jet velocity was slightly higher than the upwash velocity.

The main cause of the skin friction with embedded hairpin vortices is the vorticity increase associated with the stretching legs, which bring continuously high velocity to the wall. Figure 10 compares the circulation around the hairpin structure with and without control. To determine the effect of the control jet in reducing the skin friction of the hairpin vortex, the circulation was computed by integrating the streamwise vorticity, $\omega_x = (\frac{\partial \omega}{\partial y} - \frac{\partial v}{\partial z})$. Due to the fact that the streamwise vorticity

$\omega_x$ is proportional to $\partial u / \partial z$, the total circulation $\Gamma_x$ around the hairpin vortex structure was calculated by integrating the streamwise vorticity $\omega_x$ over the cross-sectional area. The control jet targeted the neck and head part of hairpin vortex at $225 \leq x+ \leq 300$ and suppressed the circulation around them. As the head detaches from the rest of the vortical structure, the stretching and intensifying process of the legs is reduced at $45 \leq x+ \leq 200$. In other words, the reduction in the circulation leads to a decrease in the downwash of high-momentum fluid outside the legs. The circulation relating to the wallward momentum of induced flow by the legs was reduced by up to 60%.

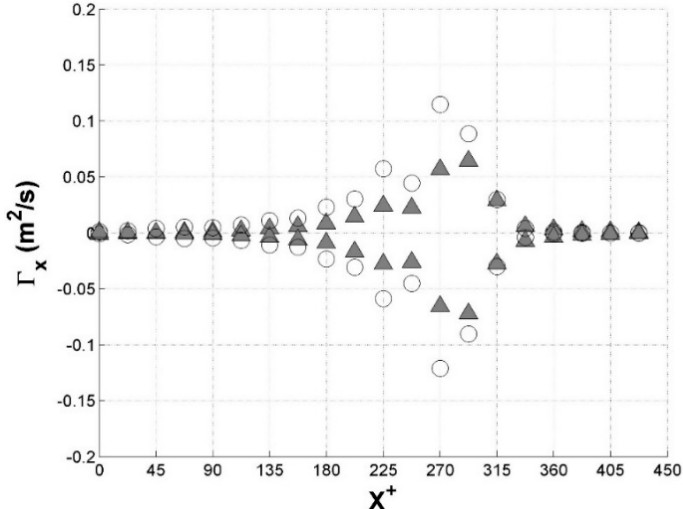

**Figure 10.** Comparison of the circulation along the streamwise direction. No control (open circle) and control (closed triangle) cases.

## 4. Conclusions

The first objective of this study was to create a single hairpin vortex in a laminar boundary layer. To do so, a vortex ring issuing through a hole on a flat plate was used to generate a hairpin vortex. In the flow visualization results, once a vortex ring pulsed into the cross flow, it developed a hairpin vortex structure. Next, a direct control strategy was applied to target the hairpin vortex with a view to reducing the skin friction. It was observed in both the flow visualization and hot-film measurement that the control jet from a nozzle disconnected the link between the vortex head and its legs, which diminished the velocity fluctuations. This meant that the control jet disconnected the head from the vortex shoulders and detached it from the rest of the vortical structure. As a result, the circulation around the legs was reduced. This, in turn, decreased the downwash of high-momentum fluid outside the legs.

Based on Taylor's frozen-flow hypothesis and the marker function, the ensemble averaged velocity fluctuations revealed a three-dimensional hairpin vortex structure between the positive and negative velocity fluctuations. This enabled a demonstration of how the control jet effectively destroyed the hairpin vortex structure. A comparison of the circulations was made, based on the streamwise vortex $\omega_x$, which showed that the circulation by the legs was reduced by up to 60%. We speculate that the weakened vorticity caused the reduction in skin friction by decreasing the high momentum toward the wall. The total skin-friction reduction could be adjusted depending on the frequency with which the hairpin vortices were targeted. If the hairpin vortices were generated and controlled more frequently, it would be expected that the reduction in the friction drag could be increased. In addition, if the control jet was exactly matched with the upwash velocity, greater reduction would be expected. This study was conducted as a preliminary work for an investigation of large-scale turbulent-boundary-layer control.

**Author Contributions:** Conceptualization, Y.-D.K.; methodology, B.K. and Y.-D.K.; software, B.K. and Y.-D.K.; validation, Y.-D.K.; data curation, B.K. and Y.-D.K.; writing—original draft preparation, B.K.; writing—review and editing, Y.-D.K.; visualization, Y.-D.K. All authors have read and agreed to the published version of the manuscript.

**Funding:** This research received no external funding.

**Conflicts of Interest:** The author declares no conflict of interest.

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
