# Peer review of "Control of Synthetic Hairpin Vortices in Laminar Boundary Layer for Skin-Friction Reduction"

_jmse, doi:10.3390/jmse8010045_

Round 1
Reviewer 1 Report
This manuscript attempts to control the developing/stretching hairpin vortex by directly injecting the jet to the head of the hairpin vortex. The final goal of the present study is to control the large scale turbulent boundary layer control represented by ships' bottom to reduce frictional resistance.
The topic reported herein is very interesting to the reviewer. Review of relevant literatures is likely to be sufficient enough. The description of measurement system as well as its configurational validity (e.g. generated laminar boundary layer profile matches to Blasius solution, generation of hairpin vortex) are well written.
Measurement results are also well reported, however, it is still unclear for the reviewer how the present results contribute to control the turbulent boundary layer (and finally, reduce frictional resistance). The mechanism of reducing high velocity gradient around the leg of the hairpin vortex by jet injection which results in reducing friction seems scientifically reasonable, however, the present results/mechanism is under laminar boundary layer with certain scale (or strength) of hairpin vortex.
For ships' bottom (assuming large side tangential and flat bottom), the boundary layer is fully turbulent. There are several scales of hairpin vortices everywhere. As long as the present method is an "active" control of hairpin vortices, what will be an idea to detect hairpin vortex to be controlled at ships' bottom? Please comment and clarify it in text. In short, the scale of the present physics and the final goal the authors stated at introduction is largely deviated which makes readers (and the reviewer) confused.
Otherwise, the manuscript is well written.
Reviewer 2 Report
Lines 82-83: This statement contradicts the statement at lines 65 and 66 which says that skin friction was increased when high-speed fluid was transported to the wall.
Line 118: How do you control leading edge separation by attaching a flap at the trailing edge? This does not make much sense.
Line 135: This statement completely destroys the accuracy of your experiment. Please check the numbers.
Line 168: I suggest maintain consistency. In a previous instance of this article, the authors has stated that the jets are located 40 cm (Line 137) downstream of the leading edge.
Line 296: It is not clear to me how the authors simplified the three components of vorticity vector to what the shown here. I suggest explaining this clearer than this.
Lines 306: The explanation for figure 10 is not rigorous. It is good if authors can explain the reasons for changing circulation at different streamwise locations of the flow.
Line 324: I think some work have been done earlier with regard to this statement. See J. Fluid Mech., 175, pp 43-83 and J. Fluid Mech., 175, pp 1–41.
Reviewer 3 Report
Please see comments in attached PDF.

Round 2
Reviewer 1 Report
The present version seems acceptable for publication.